# An Indeterminate for Malignancy FNA Report Does Not Increase the Surgical Risk of Incidental Thyroid Carcinoma

**DOI:** 10.3390/cancers14215427

**Published:** 2022-11-03

**Authors:** Davide Seminati, Eltjona Mane, Stefano Ceola, Gabriele Casati, Pietro Putignano, Mattia Garancini, Andrea Gatti, Davide Leni, Angela Ida Pincelli, Nicola Fusco, Vincenzo L’Imperio, Fabio Pagni

**Affiliations:** 1Department of Pathology, University of Milano—Bicocca (UNIMIB), 20900 Monza, Italy; 2Endocrinology, ASST Monza, San Gerardo Hospital, 20900 Monza, Italy; 3Surgery, ASST Monza, San Gerardo Hospital, 20900 Monza, Italy; 4Radiology ASST Monza, San Gerardo Hospital, 20900 Monza, Italy; 5Division of Pathology, IEO, European Institute of Oncology IRCCS, 20141 Milan, Italy

**Keywords:** incidental thyroid carcinoma, papillary thyroid carcinoma, fine needle aspiration

## Abstract

**Simple Summary:**

Incidental thyroid carcinomas (ITCs) are a frequent finding, and papillary thyroid microcarcinoma is generally the most frequent entity. We analyzed ITCs found in thyroid samples from patients that underwent surgery for other cytologically indeterminate (Bethesda classes III and IV) but histologically benign nodules to evaluate the impact of these ITCs in the expected risk of malignancy of the current cytological classes.

**Abstract:**

Incidental thyroid carcinomas (ITCs) are a fairly frequent finding in daily routine practice, with papillary thyroid microcarcinoma being the most frequent entity. In our work, we isolated incidental cases arising in thyroids removed for other cytologically indeterminate and histologically benign nodules. We retrospectively retrieved cases with available thyroid Fine Needle Aspiration (FNA, 3270 cases), selecting those with an indeterminate cytological diagnosis (Bethesda classes III–IV, 652 cases). Subsequently, we restricted the analysis to surgically treated patients (163 cases) finding an incidental thyroid carcinoma in 22 of them. We found a 13.5% ITC rate, with ITCs representing 46.8% of all cancer histologically diagnosed in this indeterminate setting. Patients received a cytological diagnosis of Bethesda class III and IV in 41% and 59% of cases, respectively. All ITC cases turned out to be papillary thyroid microcarcinomas; 36% of cases were multifocal, with foci bilaterally detected in 50% of cases. We found an overall ITC rate concordant with the literature and with our previous findings. The assignment of an indeterminate category to FNA did not increase the risk of ITCs in our cohort. Rather, a strong statistical significance (*p* < 0.01) was found comparing the larger size of nodules that underwent FNA and the smaller size of their corresponding ITC nodule.

## 1. Introduction

The improvement of high-resolution ultrasonography (US)-guided Fine Needle Aspiration (FNA) as a first-level diagnostic tool in thyroid pathology has increased the detection of particularly small lesions, producing a significant number of unexpected malignant reports [1,2,3,4,5,6,7]. Incidental thyroid carcinomas (ITCs) are defined as clinically unsuspected malignant lesions, histologically distinct from the surrounding parenchyma and found serendipitously after surgery for benign diseases [2,3,8,9,10,11].

In our previous experience, ITCs accounted for 45% of patients with a papillary thyroid carcinoma (PTC) diagnosis [5]. These results were influenced either by the small dimensions of these nodules, often below the 5 mm threshold of the US technique, or less frequently by misinterpretation errors of the obtained FNA. In this setting, ITCs arising in a multinodular goiter background, as well as follicular variants, can be particularly challenging due to the presence of these underlying confounding factors, hampering the pre-operative detection of ITCs. Further analysis of the correlation of ITCs with preoperative FNA results, as well as with comprehensive clinical and US data, can help to shed light on the risk stratification capabilities of the cytopathologist armamentarium in thyroid cancer. For this reason, the present study evaluated the demographic and clinical features of ITCs from a cohort of cases that underwent thyroid surgery for other cytologically indeterminate (Bethesda System for Reporting Thyroid Cytopathology—BSRTC categories III–IV) but histologically benign nodules. This may help in a better comprehension of the role of BSRTC classes III–IV in predicting the presence of ITCs as compared to the general population, eventually prompting a revision of these classes’ expected risk of malignancy rate within the cytological classifications.

## 2. Materials and Methods

We retrospectively collected data on patients who underwent FNA at the ASST Monza, University of Milano-Bicocca (Italy) from 1 January 2018 to 31 July 2022. From the initial cohort, we selected cases with a final cytological indeterminate diagnosis (BSRTC III: atypia of undetermined significance or follicular lesion of undetermined significance, AUS/FLUS; or IV: follicular neoplasm or suspicious for follicular neoplasm, SN/SFN) [12]. FNAs were performed by a pathologist with more than ten years of experience in cytopathology under US guidance, with 22–25-gauge needles attached to a 20 mL syringe, as previously described [13,14]. The aspirated material was smeared onto 3–4 traditional slides per nodule. The slides were then fixed with spray alcohol (Cytofix, propan-2-ol) and thus stained with Papanicolau, or air dried and stained with May–Grunwald–Giemsa. Of these, cases that underwent surgery and with available histological diagnosis were finally selected for the study. Histological evaluation was performed on surgical specimens of total or hemi-thyroidectomy, carried out according to the Italian consensus guidelines [15]. Surgical specimens were weighed, measured, inked, and evaluated from 5 mm thick anatomic slices. The tissue was formalin fixed, paraffin embedded (FFPE), and hematoxylin and eosin (H&E) stained. All the specimens were reviewed by two experienced pathologists, and the final diagnoses were classified according to the World Health Organization classification of thyroid neoplasms (2017) and staged according to the American Joint Committee on Cancer (AJCC, 8th edition) guidelines [16,17]. ITC was defined as an asymptomatic thyroid malignant lesion not identified preoperatively by cytological assessment and diagnosed on final histopathological examination of the surgical thyroid specimen. At the histological examination, nodules that underwent FNA were recognized through a comprehensive comparison with the radiological and pathological information (location and diameter), as well as by the detection of biopsy site changes caused by the FNA procedure. All thyroid cancer patients’ medical records were reviewed for demographic information (age and sex) and clinical data (previous diagnosis of Hashimoto’s thyroiditis or Graves’ disease), and for BSRTC class and histopathological characteristics, including kind of surgery, thyroid weight (if a total thyroidectomy was performed), histotype, mean tumor size, number of cancer foci, dominant tumor site, presence/absence of capsular and vascular invasion, extrathyroidal extension, complete/non-complete resection, presence/absence of metastasis to lymph nodes and to distant organs, and tumor stage. Multifocality was defined when at least two cancer foci were detected in one (unilateral) or both (bilateral) lobes; in multifocal cases, the dimension of the largest cancer nodule was used for statistical analysis. Statistical analysis was performed using the open-source R software v.3.6.0. The numbers and percentages were calculated for qualitative variables, while the means and standard deviations were calculated for quantitative variables. Student’s t-test was employed for quantitative variables. All *p* values were two-sided, and a level of <0.05 was considered to be statistically significant. This study was conducted according to the guidelines of the Declaration of Helsinki and approved by the ASST Monza Ethical Board (FINAL-TIR PU 3581/21); appropriate informed consent was obtained from all patients.

## 3. Results

The flowchart with the inclusion criteria is illustrated in Figure 1.

In the study period, 3270 patients underwent thyroid FNA, and 652 (19.9%) of them received an indeterminate for malignancy report, the general features of which are summarized in Table 1.

Thyroid surgery was performed in 163/652 (25%) indeterminate patients: 99/163 (60.7%) patients had a Bethesda category IV, while 64/163 (39.3%) had a Bethesda category III. BSRTC III patients were operated on due to clinical reasons like unaesthetic nodules or compressive symptoms (31/64, 48.5%) or for multinodularity at US examination (33/64, 51.5%), while a follow-up by repeated FNA or serial US-FNA was planned for the remaining patients according to the current guidelines [12,18,19]. Total thyroidectomy was performed in 133/163 (81.5%) cases, and surgical specimens had a 24 g mean weight (from 12 g to 56 g).

Thyroid cancer was diagnosed in 47/163 (29.9%) patients: 22/47 (46.8%) of those were ITCs. The ITC series included patients with an average age at time of surgery of 63 (48–82) years old, mainly female (72.7%, 16/22) (Table 2 and Appendix A).

Incidental cancer patients underwent surgery for cytological diagnosis of Bethesda classes III and IV in 9/22 (40.9%) and 13/22 (59.1%) cases, respectively. Total thyroidectomy was performed in 95.4% (21/22) of patients, and in class III cases, it was performed due to a symptomatic/unaesthetic nodule or due to multinodularity in 55.5% (5/9) and 44.5% (4/9) of cases, respectively. Hashimoto’s thyroiditis was present in 3/22 (13.6%) cases, while no patient had Graves’ disease. Cancers had a mean tumor size of 3,7 mm (standard deviation 2,7 mm), with carcinomas less than 10 mm and 5 mm in diameter representing 100% and 85% (17/20) of cases, respectively.

Histologically, ITCs were papillary thyroid carcinoma (PTC) conventional variants and PTC follicular variants in 50% (11/22) and 32% (7/22) of cases, respectively, while 18% (4/22) of cases had both variants. In total, 36% (8/22) of cases were multifocal, with foci bilaterally detected in 50% (4/8) of cases. The site of the dominant carcinoma was the left lobe in 55% (12/22). Regarding the cancer’s location with respect to the nodule that received the indeterminate FNA, ITC was ipsilateral, contralateral, or bilateral in 50% (11/22), 31.8% (7/22), and 18.2% (4/22), respectively. Almost no ITC had capsular invasion (4.5%, 1/22), and no case had vascular invasion, extrathyroidal extension, or non-complete resection. No lymph node metastases or metastases to distant organs were noted; therefore, according to the American Joint Committee on Cancer (AJCC, 8th edition) tumor staging, all cases were classified into stage I [17].

## 4. Discussion

Incidental thyroid nodules measuring at least 5 mm are found on 19–67% of US neck examinations, with an US malignancy risk of 1.6–29% [11,20,21,22,23,24,25]. In the last years, a significant increase in these imaging-detected incidental nodules has occurred, although the proportion of incidental and clinically known cancers has remained stable [1,2,3,5,7,26]. The malignancy risk of incidental nodules seems to decrease 2.2% per year from age 20 to 60, but cancers identified in older patients are prone to higher-risk histologies [27]. The incidence of ITCs among patients undergoing surgery for benign thyroid diseases (mostly multinodular goiter and thyroiditis) ranges between 2% and 44%. This is largely due to different patient selection criteria and variable dimensional cut-offs for FNA, often resulting in the underestimation of microcarcinomas (i.e., malignant lesions measuring <1 cm in diameter, according to the WHO Classification of Endocrine Tumours, 2017) [2,5,16,20,21,28,29]. In addition, an underlying multinodular or thyroiditis background may hamper the detection of these subtle microcarcinomas via FNA, leading some authors to propose alternative complementary approaches (e.g., elastosonography) [5,30,31,32]. As a result, most pathological ITCs reported in the literature are, indeed, microcarcinomas, typically of the papillary histotype (PTMCs), representing about half of all diagnosed PTCs [9,29,33,34].

In previous experiences, the incidence of ITCs not detected by imaging ranged from 2.3 to 13%, with differences partly due to heterogeneity in terms of iodine supply and US-FNA diagnostic performance among different regions [3,22]. In our series, we collected 652 patients with an indeterminate cytological diagnosis (19.9%, 652/3270 vs. 29.6% in a recent Italian review), classified into BSRTC categories III and IV at 13.8% (451/3270) and 6.1% (201/3270) [35]. Patients who underwent surgery accounted for 25% (163/652), with lower surgery rates for class III and IV cases as compared to the literature (14.2%, 64/451 and 49.2%, 99/201, respectively) [35]. These findings can be explained by the recent clinical trend of implementing a more conservative approach by a follow-up of repeated FNA or molecular analyses, especially for indeterminate lesions with slow growth velocity and relatively small size (our series average diameter: 22.5 mm). Thyroid cancer was detected in 28.8% of cases (47/163), compared with the overmentioned review data of 32.4% [35]. Starting from 163 patients who underwent surgery for a cytological indeterminate diagnosis and selecting only cases with indeterminate nodules that subsequently proved to be benign on histologic examination, we found 22 ITCs (13,5%, 22/163), and all carcinomas turned out to be PTMCs, as already reported in the literature (Appendix A) [1,2,3,22].

Moreover, this experience demonstrated a similar incidence of ITCs in the subset of cytological indeterminate thyroid nodules (46.8%, 22/47) as compared to the general population (45.2%, 127/281), suggesting that the coexistence of a class III/IV nodule does not increase the risk of having an ITC after surgery. Similarly, Chen et al. found 41.2% of ITCs among BSRTC category III nodules [36]. Nonetheless, the evidence-based cancer risk in cytological categories III–IV turned out to be higher as compared to the expected risk rates reported in the BSRTC system (26.5% vs. 5–15% for class III and 31.3% vs. 15–30% for class IV) [12]. The wrong radiological choice of FNA target was reasonably the main reason for ITCs, probably due to the larger diameters of nodules which underwent FNA (mean size 24 mm vs. 3.7 mm, *p* < 0.01), prompting the adoption of more efficient US screening algorithms, as previously proposed [5,13,14,37]. Moreover, we did not find a higher ITC risk in class IV cases compared with class III ones. Concerning the demographic features, we found female sex to be prevalent (72.7%), and most patients (68.1%) were aged >55 years (per AJCC age groups), in accordance with the literature (Appendix A). Provenzale et al. found that incidental PTMCs were multifocal in 76.1% of cases and that they were significantly smaller (4 vs. 9 mm) compared to non-incidental PTMCs, while Chen et al. found a 34.4% multifocality rate in incidental PTMCs [36,38]. Evranos et al. analyzed 900 final histopathological diagnoses of thyroid cancer, comparing patients with incidental and non-incidental carcinomas, finding in the former group significant lower cancer multifocality [2]. In our series, 36.4% (8/22) of cases were multifocal: interestingly, 50% (4/8) displayed a combination of conventional and follicular variants (Figure 2).

In the PTMC subset, Vasileiadis et al. and Chen et al. detected bilateral cancer foci in 19.4% and 28.1%, respectively, similar to the value found in our study (18.2%, 4/22) [29,36]. Finally, larger PTMCs seem more aggressive; hence, different diameters have been proposed as cut-off thresholds to distinguish different clinical courses [9,39,40]. In any case, because of their marked prognostic heterogeneity, in the new 5th edition (2022) of the WHO classification of thyroid neoplasms, the papillary thyroid microcarcinoma is no longer recommended as a distinct pathological PTC subtype [41]. Further, PTC early diagnosis does not seem to improve overall survival, and most authors propose a conservative approach (follow-up with a twice-yearly neck US) instead of surgery, even if others consider PTMC as a sole entity with larger thyroid cancers and treat them as such [3,25,42,43,44]. A middle approach may be represented by lobectomy, performed for incidental tumors at a stage below pT1b (i.e., <1 cm in greatest dimension and limited to the thyroid, according to the AJCC 8th edition) or else for unifocal small (<1.5 cm) tumors with no evidence of extrathyroidal extension or lymph node metastases, in patients without a personal history of neck irradiation or family history of thyroid cancer and who are unlikely to undergo postoperative radioactive iodine therapy [8,17,45]. Interestingly, Kaliszewski et al. found comparable ITC prevalence rates between patients who underwent radical and subtotal surgery, while Chen et al. found that lobectomy is related to a 9.9% ITC missing rate [21,36]. In our experience, 95.4% (21/22) of patients underwent total thyroidectomy, and performing hemi-thyroidectomy would lead to 50% missed cancer diagnoses (31.8% due to a cancer nodule in the other lobe and 18.2% due to bilateral cancer foci). To date, none of our 22 ITC patients have recurred or developed metastasis. Finally, comparing the 22 ITC patients with the other 141 Bethesda class III–IV operated cases, or even with all the Bethesda class III–IV nodules (652 cases), we did not find statistically significant differences by age, sex, or size of nodules submitted to FNA. The main limitations of our work are: the retrospective design of the study, single-center patient recruitment, risk of selection bias, and possible false-negative cases for particularly small carcinomas due to the 5 mm thick anatomic slices used for histological evaluation. Finally, it should be stated that some studies (Appendix A) were published before the 4th edition (2017) of the WHO classification of thyroid neoplasms; thus, the PTC incidence may be slightly overestimated due to the addition of NIFTPs (Non-Invasive Follicular Thyroid Neoplasm with Papillary-like nuclear features) [16,46].

## 5. Conclusions

We did not find a specific higher ITC risk in cases undergoing surgery for cytological indeterminate but histologically benign nodules as compared to the general population or in Bethesda class III versus class IV cases. The addition of ancillary techniques and the improvement of existing US screening approaches will probably help in the future to reduce the rate of ITCs and increase their preoperative detection. Rather, strong statistical significance (*p* < 0.01) was found when comparing the larger size of nodules that underwent FNA and the smaller size of their corresponding ITC nodules.

## Figures and Tables

**Figure 1 cancers-14-05427-f001:**
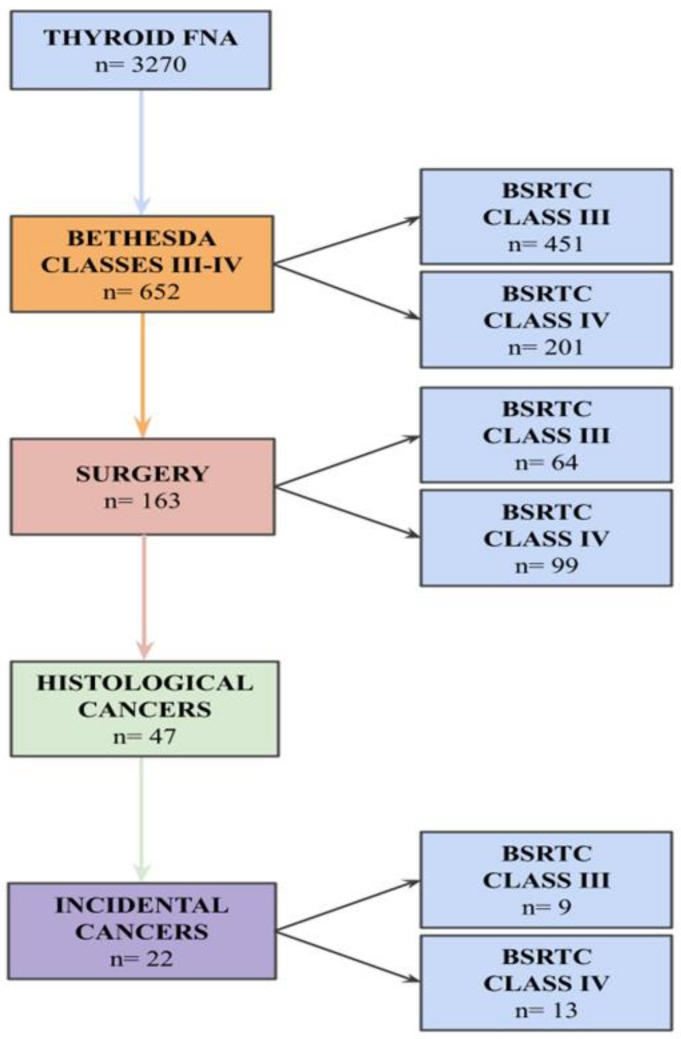
Flowchart for the selection of patients (FNA: Fine Needle Aspiration; BSRTC: Bethesda System for Reporting Thyroid Cytopathology).

**Figure 2 cancers-14-05427-f002:**
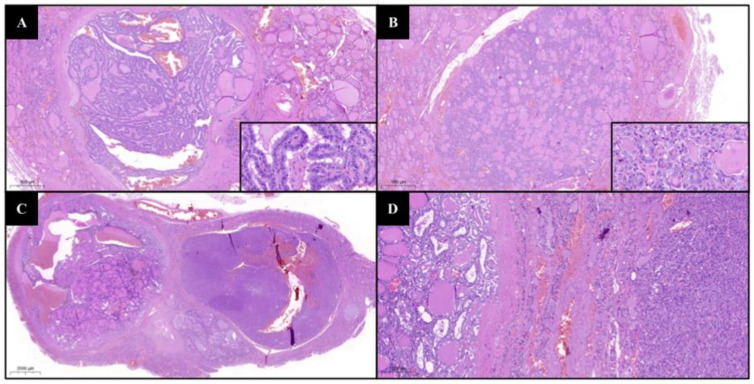
(**A**,**B**) Multifocal PTMC simultaneously presenting cancer foci with conventional papillary (**A**) and follicular (**B**) variants (H&E, 3.5× insets at 57×); (**C**,**D**) conventional variant of PTMC adjacent to the nodule submitted to FNA with cytological Bethesda class IV diagnosis and with a final histological diagnosis of oncocytic adenoma (H&E, 0.8× and 9×).

**Table 1 cancers-14-05427-t001:** Features of patients with cytologically indeterminate (Bethesda classes III–IV) nodules (AJCC: American Joint Committee on Cancer; FNA: Fine Needle Aspiration).

Classes III–IV Patient Parameters		Results
**Number of Patients**		652
AJCC AGE GROUPS (years)	<55	39.8% (260)
	≥55	60.2% (392)
SEX	Male	30.8% (201)
	Female	69.2% (451)
FNA NODULE SIZE (mm)	≤10	17% (111)
	11–20	40.9% (267)
	21–30	21.9% (143)
	31–40	13% (85)
	>40	7% (46)
FNA RESULT	Bethesda III	69.2% (451)
	Bethesda IV	30.8% (201)
SURGICAL RESECTION	Yes	25% (163)
	No	75% (489)
BETHESDA CLASS III SURGICAL REASON	Symptomatic or unaesthetic	48.4% (31)
	Multinodularity	51.5% (33)

**Table 2 cancers-14-05427-t002:** Demographic and clinico-pathological characteristics of ITC patients (AJCC: American Joint Committee on Cancer; SD: standard deviation; FNA: Fine Needle Aspiration; PTC: papillary thyroid carcinoma).

ITC Patient Parameters		Results
**Number of Patients**		22
AJCC AGE GROUPS (years)	<55	31.9% (7)
	≥55	68.1% (15)
SEX	Male	27.3% (6)
	Female	72.7% (16)
HASHIMOTO’S THYROIDITIS	Present	13.6% (3)
	Absent	86.3% (19)
GRAVES’ DISEASE	Present	0% (0)
	Absent	100% (22)
FNA NODULE SIZE (mm)	≤10	27.2% (6)
	11–20	22.7% (5)
	21–30	18.2% (4)
	31–40	31.8% (7)
	>40	0% (0)
MEAN NODULE SIZE (mm)	24 (SD: 11.6)	
FNAB RESULT	Bethesda III	40.9% (9)
	Bethesda IV	59.1% (13)
HISTOLOGY	Papillary (PTC)	100% (22)
	Follicular	0% (0)
PTC HISTOLOGICAL SUBTYPE	Conventional	50% (11)
	Follicular	31.8% (7)
	Conventional and follicular	18.2% (4)
DOMINANT SITE	Right lobe	45.4% (10)
	Left lobe	55.6% (12)
TUMOR SIZE (mm)	≤10	100% (22)
	>10	0% (0)
MEAN TUMOR SIZE (mm)	3.7 (SD: 2.7)	
CANCER FOCI	1	63.6% (14)
	≥2	36.4% (8)
CAPSULAR INVASION	Present	4.5% (1)
	Absent	95.4% (21)
VASCULAR INVASION	Present	0% (0)
	Absent	100% (22)
EXTRATHYROIDAL EXTENSION	Present	0% (0)
	Absent	100% (22)
RESECTION	Complete	100% (22)
	Non-complete	0% (0)
LYMPH NODE METASTASIS	Present	0% (0)
	Absent	100% (22)
DISTANT METASTASIS	Present	0% (0)
	Absent	100% (22)
AJCC TUMOR STAGE	I	100% (22)
	II	0% (0)
	III	
	IV	

## Data Availability

The data presented in this study are available on request from the corresponding author.

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
