# Peer review of "An Indeterminate for Malignancy FNA Report Does Not Increase the Surgical Risk of Incidental Thyroid Carcinoma"

_cancers, 2022, doi:10.3390/cancers14215427_

Round 1

Reviewer 1 Report

The authors present the retrospective study concerns the extremely important clinical problem i.e. indeterminate cytological diagnoses of thyroid nodules. The paper is well written, the study design is appropriate to answer the research question and the conclusions are supported by the presented evidence. However, I would like to indicate some minor and major issues, which should be discussed, clarify, specify and correct. The issues are as follow:

#Incidental thyroid carcinoma can be palpable before histopathological examination – please correct

#We do not have something like “radiological ITC” we can not establish diagnosis of cancer in any organ (and also in thyroid) on the basis of radiological findings – please correct

#Please do not write “the aim of this new study …” – it is funny, because it means that you can also say “the aim of this old study …” – please correct

#please do not distinguish “the pathological” and some other ITC (like radiological) - it is not correct, we do not have any other ITC, we have only just ITC (it is obvious that all of them are pathologically confirmed) – please correct

#category III of TBSRTC in not defined as “indeterminate for malignancy, low risk” it is “atypia of undetermined significance or follicular lesion of undetermined significance” and category IV is not “indeterminate for malignancy, high risk” but it is “follicular neoplasm or suspicious for follicular neoplasm” – please correct

#ITC can not be clearly identified preoperatively by laboratory tests, imaging analyses” – please correct

# what does it mean “Grave’s disease”?, do you mean Graves disease? – please correct

#Figure 1. flowchart for … - please use capital letter in the first word

#Table 1. Features of patients … - see above

#the authors said that they used 8th Edition of AJCC, so in this edition cut off the end point of age is not 45 but 55 years old, why the authors used 45 cut off – please correct or specify

#the authors should clearly explain how they recognized ITC in patients with III-IV category, it means that one nodule was biopsied and received III-IV category, and in some other nodule the pathologist recognized the cancer? How was it described, how we can be sure that in some other nodule, not biopsied, the cancer was established? – please clarify

#we should not use the term “gender” but rather “sex”, gender is rather for social description

#in table 2. what does it mean : Grave’s disease?

# line 157 page 5 please correct “the the left lobe”

# line 270 “below pT1b (i.e. < 1 cm” it shouldn’t rather be “≤  1 cm …” please check

#what does it mean “controlater cancer” line 281 page 8 please check and correct

#line 300 page 8: before 4th edition 2017 of the WHO classification? – please check

Reviewer 2 Report

An Indeterminate for Malignancy FNA Report does not increase the Surgical Risk of Incidental Thyroid Carcinoma

Davide Seminati 1,*, Eltjona Mane 1 , Stefano Ceola 1 , Gabriele Casati 1 , Pietro Putignano 2 5 , Mattia Garancini 3 , Andrea Gatti 3 , Davide Leni 4 , Angela Ida Pincelli 2 , Nicola Fusco 5 6 , Vincenzo L’Imperio 1 and Fabio Pagni 1

I appreciate the work on this study, but in general it is very confusing and incorrect. 

There are just some comments. 

Incidental thyroid carcinoma (ITC) was defined by the authors as an asymptomatic thyroid malignant lesion not clearly identified preoperatively by laboratory tests, imaging analyses or cytological assessment and diagnosed on final histopathological examination of the surgical thyroid specimen. But there is a clear definition of so-called “incidentaloma”. 

A thyroid incidentaloma is defined as an unexpected, asymptomatic thyroid tumor (benign or malignant) discovered during the investigation of an unrelated condition. Further, in Table 2 is the characterization of the ITC group and there are mentioned FNA results? It is very common that for Bethesda III+IV the malignancy risk is around 30%. I don´t understand what exactly you mean with the incidental TC? The next definition of thyroid incidentaloma is when you send the patient for thyroid surgery for example for benign disease and incidentally there is a small usually PTC (“incidentaloma”). 

Abstract: Rather, a strong statistical significance (p < 0.0001) was found comparing the size of nodules that underwent FNA and the size of their corrispective ITC nodule. 

Statistical significance means bigger or smaller?

BSRTC III (indeterminate for malignancy, low risk) it should be AUS/FLUS or IV (indeterminate for malignancy, high risk) it should be FN/SFN, this is the official description. 

5th edition of WHO classification of thyroid neoplasms the papillary thyroid microcarcinoma is no longer recommended as a distinct PTC subtype. This is right, but this is pathological classification and not clinical. 

In general, English must be improved, it is hard to read it. It must be clear what the study is about. 

Round 2

Reviewer 2 Report

I still find the study very hardly readable. English must still be improved. I think that there are many misunderstandings because of poor English. But the major weakness of this study is the definition of incidentaloma line 42-48 and further the line 223-228…

Comment 1: You use FNA results Bethesda III and IV, so already patients with some cancer suspicion and I could not find anywhere in your methods how did you identify the thyroid nodule undergoing FNA in comparison to histology, is it the same nodule or is it not? How do you know it for sure? 

Comment 2: line 152-161 

Incidental cancer patients underwent surgery for cytological diagnosis of Bethesda class IV and III in 13/22 (59.1%) and 9/22 (40.9%) cases, respectively…..Cancers had a mean tumor size of 3,7 mm (standard deviation 2,7 mm), with carcinomas less than 10 mm and 5 mm in diameter representing 100% and 85% (17/20) cases, respectively. 

In comparison to Table 2. Demographic and clinico-pathological characteristics of ITC patients, there are different thyroid nodule sizes.

Comment Abstract 3: We retrospectively selected patients who underwent Fine Needle Aspiration from 01/01/2018 to 31/07/2022 (3270 cases)

-the dates belong to methods and not abstract 

p < 0.000

-what about to make shorter form p < 0.01

Rather, a strong statistical significance (p < 0.0001) was found comparing the size of nodules that underwent FNA and the size of their corresponding ITC nodule. 

You should clearly write it. From this sentence I just know that there was significance, it is not enough. 

In general, discussion is too long.
